# Importance of Fecal Microbiota Transplantation and Molecular Regulation as Therapeutic Strategies in Inflammatory Bowel Diseases

**DOI:** 10.3390/nu16244411

**Published:** 2024-12-23

**Authors:** Olga Brusnic, Adrian Boicean, Sorin-Radu Fleacă, Blanca Grama, Florin Sofonea, Corina Roman-Filip, Iulian Roman-Filip, Adelaida Solomon, Sabrina Birsan, Horatiu Dura, Corina Porr, Cristian Adrian, Danusia Maria Onisor

**Affiliations:** 1Department of Internal Medicine VII, George Emil Palade University of Medicine, Pharmacy, Science and Technology of Targu Mures, Gheorghe Marinescu Street No. 38, 540136 Targu Mures, Romania; brusnic_olga@yahoo.com (B.O.); halalisan5@yahoo.com (D.M.O.); 2Faculty of Medicine, Lucian Blaga University of Sibiu, 550169 Sibiu, Romania; radu.fleaca@ulbsibiu.ro (S.-R.F.); corina.roman@ulbsibiu.ro (R.-F.C.); solomonadelaida@gmail.com (S.A.); sabrina.marinca@yahoo.com (B.S.); horatiu.dura@ulbsibiu.ro (H.D.); corina_sibiu@yahoo.com (P.C.); adrian.cristian@ulbsibiu.ro (A.C.); 3Faculty of Social Sciences, Lucian Blaga University of Sibiu, 550012 Sibiu, Romania; blanca.grama@ulbsibiu.ro (G.B.); florin.sofonea@ulbsibiu.ro (S.F.); 4Department of Neurology, “George Emil Palade” University of Medicine, Pharmacy, Sciences and Technology, 540136 Targu Mures, Romania; iulian_roman2009@yahoo.com

**Keywords:** IBD, miRNA, personalized treatment

## Abstract

Noncoding RNAs, particularly microRNAs (miRNAs) and small interfering RNAs (siRNAs), have emerged as key players in the pathogenesis and therapeutic strategies for inflammatory bowel disease (IBD). MiRNAs, small endogenous RNA molecules that silence target mRNAs to regulate gene expression, are closely linked to immune responses and inflammatory pathways in IBD. Notably, miR-21, miR-146a, and miR-155 are consistently upregulated in IBD, influencing immune cell modulation, cytokine production, and the intestinal epithelial barrier. These miRNAs serve as biomarkers for disease progression and severity, as well as therapeutic targets for controlling inflammation. This comprehensive review highlights the intricate interplay between the gut microbiota, fecal microbiota transplantation (FMT), and miRNA regulation. It concludes that microbiota and FMT influence miRNA activity, presenting a promising avenue for personalized IBD treatment.

## 1. Introduction

Fecal microbiota transplantation (FMT) is an emerging area of research for its potential therapeutic benefits in inflammatory bowel diseases (IBD), such as Crohn’s disease and ulcerative colitis. Recently, there has been a growing interest in exploring FMT as a type of “vaccine” to modulate the immune system and maintain remission in IBD patients [1,2].

FMT involves transferring stool from a healthy donor into the gastrointestinal tract of a patient. This process aims to restore a balanced microbial community in the recipient’s gut, which can be disrupted in individuals with IBD. IBD is often associated with dysbiosis, an imbalance in gut microbiota. FMT helps restore microbial diversity, which can enhance the gut’s resilience against inflammation [3,4].

FMT may be considered a vaccine-like treatment line; the idea of using FMT as a “vaccine” in IBD involves several key aspects, such as microbial diversity restoration and immune modulation, and the most important result could be long-term remission [5,6,7].

FMT may represent a key therapy line and outline the concept of personalized medicine; nowadays, research focuses on personalized FMT approaches, tailoring treatments based on individual microbial profiles and specific disease characteristics. Moreover, FMT may be used in conjunction with other treatments, such as immunosuppression drugs or biologics, to enhance therapeutic outcomes [7,8,9,10,11,12,13,14,15]. The gut microbiota plays a crucial role in modulating the immune system. By introducing a healthy microbial community, FMT can potentially downregulate inflammatory responses and promote immune tolerance. Like vaccines, which aim to provide long-term protection against diseases, FMT aims to induce and maintain long-term remission in IBD patients by stabilizing the gut environment [15,16,17,18,19,20].

Several studies and clinical trials are investigating the efficacy of FMT in IBD. Early trials have shown that FMT can be safe and effective in inducing remission in some patients with IBD, particularly those with ulcerative colitis. Research is ongoing to understand the specific mechanisms through which FMT exerts its effects, including changes in microbial composition, metabolite production, and immune regulation [8,9,10]. Identifying optimal donor characteristics is crucial for the success of FMT. Donors are typically screened for a wide range of pathogens to ensure safety [20,21,22,23,24,25,26,27,28,29,30]. Efforts are being made to standardize FMT procedures, including donor screening, stool processing, and administration methods, to ensure the consistency and reproducibility of results [30,31,32,33,34,35].

FMT holds promise as a novel, personal therapy for managing IBD by restoring microbial balance and modulating the immune system. However, more research is needed to fully understand its potential and to optimize its application in clinical practice. One step towards understanding the importance of FMT in different disorders is to understand its molecular mechanisms [35,36,37,38,39,40,41,42,43,44,45,46,47]. Fecal microbiota transplantation (FMT) has intriguing implications for miRNA (microRNA) regulation, which can significantly influence the psychophysiology and treatment outcomes of various diseases, including inflammatory bowel diseases (IBD) [8,9,10]. MicroRNAs are small, noncoding RNAs that play a crucial role in gene regulation by modulating the expression of target mRNAs. Here is how FMT might affect miRNA regulation. The gut microbiota can affect the host’s miRNA expression profiles. Changes in the gut microbial composition after FMT can lead to alterations in the host’s miRNA expression, which in turn can impact various biological processes and disease states [8,9,10,47,48,49,50,51,52,53,54,55,56,57,58,59,60].

Certain bacterial species in the gut have been shown to produce metabolites that can modulate miRNA expression. For example, short-chain fatty acids (SCFAs) produced by gut bacteria can influence the expression of miRNAs involved in inflammation and immune response [11,12,13,60,61,62,63,64,65,66,67,68,69,70].

In IBD, dysbiosis is often associated with abnormal miRNA expression that promotes inflammation. FMT can restore a healthy microbial balance, potentially normalizing miRNA expression, reducing inflammation, and restoring normal miRNA levels to regulate the immune system [20,21,22,23,24,25,26,27,28,29,30,71]. MiRNAs such as miR-155 and miR-146a are known to regulate inflammatory pathways. By modulating the gut microbiota through FMT, the expression of these miRNAs can be adjusted, leading to altered inflammatory responses in the gut [11,12,13,14,15,16,20,21,22,23,24,25,26,27,28,29].

This is the first review of literature that outlines the importance of molecular changes that result after FMT in IBD and its possible clinical benefits for the treatment of IBD and establishing long remission based on molecular modifications.

## 2. Materials and Methods

In this review on the fecal microbiota transplantation (FMT) interplay with miRNA in inflammatory bowel disease (IBD), we conducted a comprehensive literature search across multiple electronic databases, including PubMed, Scopus, and Web of Science. Searches were conducted using relevant keywords such as “fecal microbiota transplantation”, “FMT”, “miRNA”, “AMPs”, and “inflammatory bowel disease” in combination with terms like “Crohn’s disease”, “ulcerative colitis”, and “gut microbiota”. Our review included studies published in English between [insert years] and prioritized randomized controlled trials, observational studies, and meta-analyses with human subjects. Studies were selected based on predefined inclusion criteria, which encompassed research on the safety, efficacy, and mechanisms of FMT in IBD. Data extraction focused on participant characteristics, FMT protocols, clinical outcomes, and microbiota-related changes.

The primary outcome of this review is to assess the importance of molecular treatments in personalized therapeutic strategies in IBD based on miRNA regulation and interconnections between miRNA and FMT. As far as we have researched in the current literature, this is the only review that outlines possible connections between miRNA and FMT in treating IBD.

### 2.1. MiRNA and AMPs Regulation in Fecal Microbiota Transplantation

Microbes in the gut can produce miRNA-like molecules that may directly interact with the host’s cellular machinery, influencing gene expression [40]. FMT can induce epigenetic changes, including DNA methylation and histone modification, which can affect miRNA expression and thereby modulate gene regulation [40,61,72]. The gut microbiota plays a crucial role in shaping the host’s immune system [71,72]. Changes in the microbial community through FMT can lead to the modulation of miRNAs that are involved in immune cell differentiation and function [72]. Clinical implications are emphasized due to changes in miRNA expression profiles after FMT, which can serve as biomarkers for treatment response, helping to predict which patients are likely to benefit from FMT [5,6,7]. Interaction between gut microbiota and miRNA expression highlights the potential for personalized medicine approaches in IBD, where treatments are tailored based on individual microbiota and miRNA profiles [73]. MiRNAs have multiple clinical benefits and are a target of new research directions to elucidate the precise mechanisms by which FMT influences miRNA expression and the downstream effects on gene regulation [11,20,21,22,23,24,25].

Investigating the long-term effects of FMT on miRNA regulation and the stability of these changes over time is crucial for understanding its potential as a durable therapeutic intervention. Exploring the synergistic effects of FMT combined with other treatments, such as miRNA-based therapies or traditional IBD medications, could enhance therapeutic outcomes [30,31,32,33,34,35,36,37,38,39,40,41,42,43,44,45].

Besides, targeted therapies such as monoclonal antibodies targeting specific cytokines or cell surface molecules (e.g., TNF inhibitors like infliximab and adalimumab, IL-12/23 inhibitors like ustekinumab) have transformed IBD treatment [71]. Genetic insights can help identify which patients are more likely to respond to these biologics; other small molecule inhibitors targeting intracellular signaling pathways (e.g., Janus kinase inhibitors like tofacitinib) based on genetic profiles can provide another layer of precision in treatment [11].

Therapies for IBD based on genetic insights are paving the way for more personalized and effective treatment strategies. By targeting specific genetic pathways and considering individual genetic profiles, these approaches hold promise for improving the outcomes and quality of life for patients with IBD [11,71]. The aim of this review is to identify potential therapeutic benefits regarding molecular therapies like miRNA and the implications of FMT in regulating the pathways and types of miRNAs and regulating antimicrobial peptides (AMPs) [73,74].

We emphasize that this is the first review of literature that describes potential personalized therapies based on FMT, miRNAs, and AMP regulation. Also, an important outcome of this review is to highlight the importance of FMT in inducing molecular changes and improving gut microbiota.

Besides miRNA, FMT may also regulate antimicrobial peptides (AMPs) in IBD. AMPs are crucial components of the innate immune system, playing a vital role in defending against pathogens and maintaining intestinal homeostasis. In IBD, the secretion and function of AMPs are often disrupted, contributing to disease pathogenesis [72,73,74]. First of all, defensins represent a major group of AMPs, with α-defensins produced by Paneth cells in the small intestine and β-defensins produced by epithelial cells; the dysregulation of AMPs is implicated in the development and progression of IBD [73,74]. In IBD, particularly Crohn’s disease, there is a notable reduction in α-defensins, leading to impaired mucosal barrier function and increased susceptibility to infections. Also, cathelicidin LL-37, the active form of cathelicidin, has antimicrobial and immunomodulatory functions; alterations in LL-37 expression have been observed in IBD patients, potentially exacerbating inflammation and barrier dysfunction [60,61,62,63,64,65,66,67,68,69,70,71,72,73,74].

Recent therapeutic approaches focus on restoring AMP levels through FMT, improving human β-defensin, which has shown promise in increasing gut microbiota diversity and improving experimental colitis in mice [72,73,74].

FMT may regulate miRNA and AMP secretion in the case of IBD, opening the pathways toward personalized molecular treatment [11,71,72,73,74].

FMT reestablishes a healthy gut microbial balance, which is crucial for the regulation of AMP production. Microbiota-derived signals can influence epithelial cells and immune cells, promoting the secretion of AMPs like defensins and LL-37 [73,74]. Additionally, FMT can modulate other pathways that indirectly affect AMP levels, such as reducing inflammation or enhancing mucosal healing, further stabilizing AMP activity [73,74,75].

Another molecular implication of FMT is represented by miRNA expression profiles, which represent potential biomarkers for treatment efficacy, enabling predictions about patient response to therapy. This aspect is particularly important for personalized medicine approaches in IBD, where interventions are tailored based on individual microbiota and miRNA signatures [73,75]. The ability to modulate miRNA through FMT underscores its therapeutic promise, particularly when combined with other targeted therapies like miRNA-based treatments or conventional IBD medications [76].

Fecal microbiota transplantation (FMT) plays a significant role in influencing host molecular pathways, particularly through its effects on microRNA (miRNA) expression and regulation in the context of inflammatory bowel disease (IBD) [72,73,74].

The gut microbiota, a critical component in maintaining intestinal and systemic homeostasis, can produce miRNA-like molecules that directly interact with the host’s cellular processes, influencing gene expression. FMT-induced changes in gut microbiota composition can lead to epigenetic modifications, including DNA methylation and histone changes, which in turn modulate miRNA profiles [71,72,75,76].

These miRNAs influence diverse cellular processes, including immune cell differentiation and function, thus shaping the host’s immune responses. Research is increasingly focused on understanding the durability of miRNA modulation following FMT and its long-term implications for gene regulation and immune function. Combining FMT with therapies such as monoclonal antibodies targeting specific cytokines (e.g., TNF inhibitors like infliximab or IL-12/23 inhibitors like ustekinumab) or small molecule inhibitors of intracellular pathways (e.g., Janus kinase inhibitors like tofacitinib) offers opportunities for synergistic therapeutic outcomes [76]. By leveraging genetic and epigenetic insights, these combinations could optimize treatment effectiveness and improve the quality of life for patients with IBD [76].

Another crucial aspect that should be further studied is represented by the long-term safety and potential immunogenicity of miRNA therapies, which need thorough evaluation in clinical trials [11,76]. Also, the development and approval of miRNA-based therapies involve rigorous regulatory oversight to ensure safety and efficacy. Table 1 highlights the interplay between antimicrobial peptides (AMPs) and microRNAs (miRNAs) in inflammatory conditions, particularly inflammatory bowel disease (IBD) [60,61,62,63,64,65,66,67,68,69,70,71,72,73,74,75,76].

Table 1 illustrates how miRNAs can either upregulate or downregulate AMP expression, thereby influencing inflammatory pathways and the immune environment in the gut. Some miRNAs (e.g., miR-21 and miR-155) tend to exacerbate inflammation, while others (e.g., miR-146a and miR-223) play more regulatory roles, balancing AMP activity and immune responses essential for gut homeostasis [71,72,73,74,75,76,77,78,79].

### 2.2. Therapeutic Strategies Based on Molecular Therapies

Current research and clinical trials concerning miRNA are developing different therapeutic strategies regarding possible ways to regulate genes in IBD, and molecular treatments in IBD could change. Various animal and human models of IBD are being used to test the efficacy of miRNA-based therapies [60,61,62,63,64,65,66,67,68,69,70,71,72,73,74,75,76,77].

These studies are essential for understanding the therapeutic potential and mechanisms of action; also, important clinical trials are focused on establishing the importance of therapeutic strategies based on miRNA; some miRNA-based therapies are progressing to early-phase clinical trials, evaluating their safety, tolerability, and preliminary efficacy in IBD patients [77].

Preclinical studies are focused on exploring specific miRNAs like miR-21, miR-146a, miR-155, and miR-223 for their roles in inflammation modulation and intestinal healing [77,78,79].

Animal models such as DSS (dextran sulfate sodium) and TNBS (2,4,6-trinitrobenzenesulfonic acid) colitis are commonly used, with miRNA delivery methods also being optimized through nanoparticle and liposomal technologies [77,78,79].

Early-phase clinical trials in humans are testing the safety, tolerability, and preliminary efficacy of these therapies, with encouraging results in symptom reduction and immune modulation observed; miRNA therapy offers a promising new avenue for the treatment of IBD by targeting specific molecular pathways involved in inflammation and immune regulation [60,61,62,63,64,65,66,67,68,69,70,71,72,73,74,75,76,77,78,79].

While still in the experimental stages, advancements in delivery technologies and a better understanding of miRNA biology could make this approach viable for IBD management [35,36,37,38,39,40,41,42,43,44,45,46,47,48,49,50,51,52,53,54,55].

In parallel, miR-146a knockout mice demonstrated resistance to the dextran sulfate sodium (DSS)-induced colitis by inhibiting genes associated with the intestinal barrier. Conversely, the overexpression of miR-146b conferred protection against DSS-induced colitis by activating NF-κB signaling and enhancing epithelial barrier function [77,78,79,80].

Based on these findings, researchers investigated the administration of miR-146a via extracellular vesicles in rats with trinitrobenzene sulfonic acid (TNBS)-induced colitis [77,78,79,80]. This approach led to increased miR-146a expression in the colon, which alleviated colitis by reducing inflammation through the MAPK and NF-κB signaling pathways. Additionally, the oral administration of miR-146b-loaded nanoparticles protected miR-146b-deficient mice from DSS-induced colitis [80].

This protective effect was characterized by a decrease in the expression of proinflammatory cytokines IL-1β and TNF-α in M1 macrophages and an increase in M2 macrophages, underscoring miR-146b’s role in promoting the transition from a proinflammatory M1 to an anti-inflammatory M2 macrophage phenotype [80]. However, conflicting data have emerged from studies showing that the suppression of miR-146a, either by a synthetic inhibitor or through the oral administration of the antidiabetic drug vildagliptin, resulted in the amelioration of experimentally induced colitis in rats [70,71,72,73,74,75,76,77,78,79,80].

A study showed that miR-146a knockout mice developed severe colitis, highlighting its protective role in intestinal inflammation. Research demonstrated that increasing miR-146a expression in mice could alleviate symptoms of experimental colitis, suggesting its therapeutic potential. An animal study found that mice deficient in miR-155 had reduced susceptibility to colitis, supporting its role in promoting inflammation [77,78,79,80].

Another study by Singh et al. showed that inhibiting miR-155 could reduce inflammatory responses in a mouse model of IBD, indicating that miR-155 antagonists might be effective in treating IBD [81]. These findings suggest that targeting miRNAs could be a promising strategy for developing new treatments for IBD. Similarly, modulating the levels of specific miRNAs like miR-146a and miR-155 could help control the chronic inflammation that characterizes this condition. However, further clinical studies are needed to translate these findings into effective therapies [77,78,79,80,81].

Based on how FMT is involved in restoring gut homeostasis and its interplay in the regulation of miRNA and AMPs, we can consider FMT a potential therapeutic tool that regulates molecular changes in miRNAs [71,72,73,74,75,76,77,78,79,80,81].

A similar and natural way to modulate the microbiota and the molecular pathways is represented by FMT. Transferring gut microbiota from a healthy donor may influence the patient’s miRNA expression. By restoring a healthy microbial balance through FMT, the expression of miRNAs involved in inflammation and immune regulation can be modulated. Gut bacteria produce metabolites such as short-chain fatty acids (SCFAs) that can affect miRNA expression. These metabolites can impact cellular processes and inflammatory responses [55,56,57,58,59,60].

Another treatment direction is regulating the disturbance in AMP secretion associated with IBD, which has opened up new perspectives of treatment with oral administration of defensins as a promising therapeutic option. Specific modifications can enrich these peptides in the mucus at different intestinal locations, protecting the epithelial layer from bacteria in the lumen. Recent findings indicate that the oral delivery of human β-defensin 2 increases gut microbiota diversity and is effective in treating experimental colitis in mice. The development of new therapeutic molecules targeting Crohn’s disease is ongoing, though clinical use is still in its early stages [77,78,79,80].

The disturbance of antimicrobial peptides (AMPs) and microRNAs (miRNAs) in Inflammatory Bowel Disease (IBD) has garnered significant interest due to their roles in maintaining gut homeostasis and immune responses [77,78]. This review primarily assessed to highlight the importance of FMT in IBD and its possible connection with molecular changes. Another outcome of the review is to outline the importance of molecular and gene therapy, recently studied, and present potential clinical benefits that could represent the future personalized treatment of IBD.

A possible genetic treatment could be RNA Interference (RNAi), which can be used to silence specific genes involved in the inflammatory response. This approach is being explored for its potential to modulate gene expression in IBD [77,78,79]. Synthetically designed miRNA mimics can restore the levels of beneficial miRNAs that are downregulated in IBD, helping to suppress inflammation as antagomirs. These are chemically modified RNA molecules designed to inhibit overexpressed, pathogenic miRNAs. For example, targeting miR-155 or miR-21, known to promote inflammation in IBD, could mitigate disease severity [77,78].

Identifying specific miRNA expression patterns in patients with IBD could guide personalized therapeutic strategies. Profiling could help classify the subtypes of IBD, predict disease progression, or monitor responses to treatment. Exploring the role of miRNAs in individual genetic predispositions and epigenetic changes could tailor more effective interventions [77,78]. Utilizing liposomes, polymer nanoparticles, or lipid nanoparticles to ensure targeted delivery to intestinal tissues while protecting miRNAs from degradation could present important clinical benefits to act as miRNA mimics; antagomirs, harnessing natural exosomes as carriers for miRNAs could enhance delivery specificity and reduce off-target effects [77,78,79,80,81]. Although these molecular therapies need further studies, highlighting their importance as potential therapeutic strategies and clinical benefits is very important and could open new research pathways; moreover, miRNA-based therapies offer a unique opportunity to modulate disease pathways with high specificity and minimal side effects. However, further research and technological advances are necessary to translate these innovations into viable treatments for IBD [77,78,79,80,81,82,83,84].

Table 2 summarizes the miRNAs that have been explored as treatments for inflammatory bowel disease (IBD). These miRNAs can influence inflammation, immune modulation, and tissue repair mechanisms, offering potential therapeutic avenues Table 3.

These miRNAs are still under preclinical and early-stage clinical investigation. Current studies emphasize targeted delivery systems to ensure they reach specific intestinal cells, as well as minimize off-target effects and improve stability within the gastrointestinal tract [71,72,73,74,75,76,77,78,79,80,81,82,83,84].

Although, at the moment, there are very few studies regarding the importance of FMT, miRNA, and AMPs in IBD, future research could be promising in establishing a personalized treatment for patients with severe forms of IBD.

## 3. Conclusions

Fecal microbiota transplantation (FMT) is emerging as a promising therapeutic approach for regulating microRNAs (miRNAs) in inflammatory bowel diseases (IBD). MiRNAs are small, noncoding RNAs essential for gene expression regulation, and their dysregulation is closely linked to IBD. This review highlights the intricate relationship between FMT, antimicrobial peptides (AMPs), and miRNAs in maintaining intestinal immunity and homeostasis. MiRNAs can modulate AMP expression, and AMPs, in turn, can influence miRNA activity, with FMT playing a pivotal role in regulating both. Understanding this interplay is critical for developing holistic therapeutic strategies for IBD. Disruptions in AMPs and miRNAs are integral to IBD pathogenesis, underscoring their interconnected roles. Notably, FMT shows potential clinical benefits by restoring miRNA and AMP balance, thereby aiding in mucosal barrier repair and immune response modulation. Targeting these molecules through therapeutic interventions offers a promising avenue for improving IBD outcomes. However, further research is needed to unravel their precise mechanisms and translate these insights into effective treatments.

## Figures and Tables

**Table 1 nutrients-16-04411-t001:** How AMPs and miRNAs regulate immune responses, inflammation, and gut homeostasis.

AMP	miRNA	Interplay Mechanism	Role in IBD
LL-37	miR-21 [77]	miR-21 modulates LL-37 expression, affecting inflammation levels	High levels of miR-21 correlate with reduced AMP function, leading to elevated inflammation
Defensins	miR-146a [78]	miR-146a controls the immune response and can upregulate defensins	Defensins help maintain intestinal barrier, while miR-146a limits excessive inflammation
HBD-2	miR-155 [78]	miR-155 affects HBD-2 expression, enhancing inflammatory response when overexpressed	Elevated miR-155 correlates with increased inflammation and impaired AMP response
RegIIIγ	miR-223 [79]	miR-223 regulates neutrophil function and maintains gut barrier by modulating RegIIIγ	Balances immune response, supporting AMP activity in intestinal integrity
Cathelicidins	miR-122 [77]	miR-122 downregulation supports cathelicidin function in inflammation	Cathelicidins protect against pathogens, while miR-122 modulation reduces fibrosis

**Table 2 nutrients-16-04411-t002:** Highlights of key miRNAs, their target genes, or pathways.

miRNA	Associated with IBD Type	Target Genes/Pathways	Role in IBD
miR-21	Crohn’s Disease (CD), Ulcerative Colitis (UC) [81,82,83]	PDCD4, TLR4, STAT3, IL-23	Promotes inflammation by enhancing proinflammatory cytokines, involved in immune response and epithelial integrity.
miR-155	CD, UC [81,82,83]	SHIP1, SOCS1	Enhances inflammation by modulating immune cell activation and cytokine production. Overexpressed in active IBD.
miR-223	CD, UC [71,81]	NLRP3, IL-1β	Regulates inflammation by modulating IL-1β and IL-6; contributes to intestinal barrier maintenance.
miR-146a	CD, UC [77,78,79,80,81]	TRAF6, IRAK1, NF-κB	Anti-inflammatory role by targeting NF-κB pathway components, regulates cytokine release in immune cells.
miR-31	CD [77]	Multiple targets, epithelial integrity	Associated with gut epithelial integrity and fibrosis regulation in the intestines. Increased in active disease.
miR-122	UC [77,78,79,80,81]	MUC1	Implicated in mucus layer maintenance; regulates epithelial cell differentiation and mucus production.
miR-124	UC [77,78,79,80,81]	STAT3, IL-6, TNF-α	Anti-inflammatory effects by targeting STAT3 and cytokine production pathways.
miR-375	UC [77,78,79,80,81]	JAK2/STAT3	Anti-inflammatory; regulates immune response and epithelial healing. Downregulated in active disease.
miR-199a	CD [77,78,79,80,81]	NF-κB pathway	Reduces intestinal inflammation by targeting NF-κB pathway, involved in immune modulation.
miR-150	CD, UC [77,78,79,80,81]	c-Myb, AKT	Regulates immune cell function, particularly T-cell responses; associated with inflammation.
miR-29a	UC [77,84]	IL-6, TNF-α, STAT3	Reduces proinflammatory cytokine production; potential biomarker for UC severity.
miR-192	UC [84]	TGF-β pathway	Anti-inflammatory, contributes to maintaining intestinal epithelial function.
miR-10a	CD, UC [77]	IL-12/IL-23, NF-κB	Modulates immune response, decreases proinflammatory cytokine production.
miR-26a	CD [77,84]	IL-6, IL-8	Downregulated in CD; linked to immune modulation and epithelial repair.

**Table 3 nutrients-16-04411-t003:** Therapeutic strategy in IBD.

miRNA	Role in IBD Pathogenesis	Therapeutic Strategy	Findings
miR-21	Upregulated in IBD, promotes inflammation via immune response regulation [77,84]	Inhibition of miR-21	Reduction in intestinal inflammation, restoration of immune balance
miR-155	Involved in proinflammatory pathways, particularly in macrophages [71,72,73,74,75,76,77,78,79,80,81,82,83,84]	Inhibition of miR-155	Decreased production of proinflammatory cytokines, improved tissue repair
miR-146a	Regulates immune response and inflammation, often downregulated in IBD [77,84]	Upregulation via mimics	Reduction in inflammation, enhanced immune regulation
miR-122	Known for its role in liver function but found relevant in gut inflammation [84]	Targeted inhibition	Reduced fibrosis, improved inflammation response

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
