# Peer review of "Importance of Fecal Microbiota Transplantation and Molecular Regulation as Therapeutic Strategies in Inflammatory Bowel Diseases"

_nutrients, 2024, doi:10.3390/nu16244411_

Round 1
Reviewer 1 Report
Comments and Suggestions for Authors
The review highlights the role of miRNAs, particularly miR-21, miR-146a, and miR-155, in IBD pathogenesis, linking them to immune regulation and inflammation. It also explores the potential of microbiota, FMT and AMP to regulate miRNAs, offering avenues for personalized IBD therapies. However, I think this manuscript needs a careful revision due to its problems of organization.
Comments:
1. The main title of this manuscript focuses on "Interplay between FMT and mi-RNA", but the authors only use a minor part to discuss the interaction between FMT and miRNA. A major part of this paper actually focuses on miRNA's functions as well as its interplay with AMP. I suggest reorganizing the manuscript including three sections: a) interplay of FMT-miRNA, b) interplay of FMT-AMP (if possible) and c) interplay of AMP-miRNA. Also, the main title of manuscript also need change.
2. Please do proofreading carefully since there are many typing errors (e.g. capital/lowercase, space).
3. Please insert the reference number behind exact site of citation instead of paragraph end.
4. Please list the references in each row of table 1-3.
5. Line 109-156: The whole paragraph discussed very little about the specific mechanism of miRNA modulation. It focuses more on introduction of current research directions, if so, I suggest changing the title.
6. Line 43-52: Please provide references.
7. Line 89: I suggest changing subtitle, this is a reviewing paper not research paper.
8. Line 108: Please unify the format of subtitles.
9. Line 217-224: Please combine with next paragraph.
10. Line 229-239: Please combine with next paragraph.
11. Line 272-287: Why did author discuss the AMPs and FMT interaction under the subtitle of "Implications of miRNAs in IBD Pathogenesis"?
12. Line 301-304: How about moving this to the end of section?
Author Response
The review highlights the role of miRNAs, particularly miR-21, miR-146a, and miR-155, in IBD pathogenesis, linking them to immune regulation and inflammation. It also explores the potential of microbiota, FMT and AMP to regulate miRNAs, offering avenues for personalized IBD therapies. However, I think this manuscript needs a careful revision due to its problems of organization.
Comments:
- The main title of this manuscript focuses on "Interplay between FMT and mi-RNA", but the authors only use a minor part to discuss the interaction between FMT and miRNA. A major part of this paper actually focuses on miRNA's functions as well as its interplay with AMP. I suggest reorganizing the manuscript including three sections: a) interplay of FMT-miRNA, b) interplay of FMT-AMP (if possible) and c) interplay of AMP-miRNA. Also, the main title of manuscript also need change.
We sincerely thank the esteemed reviewer for their valuable suggestions. We have reorganized the manuscript in accordance with your recommendations. The revised version primarily focuses on the interaction between miRNA, AMPs, and FMT in the first section, followed by a detailed discussion of current strategies centered on molecular therapies in the second section.
- Please do proofreading carefully since there are many typing errors (e.g. capital/lowercase, space).
We highly appreciate the distinguished reviewer’s observation we revised the technical details.
- Please insert the reference number behind exact site of citation instead of paragraph end.
We highly appreciate the observation in this regard and we thank again for the provided suggestions, we clarified this aspects in the manuscript.
- Please list the references in each row of table 1-3.
We highly appreciate the observation in this regard and we thank again for the provided suggestions, we clarified this aspects in the manuscript.
- Line 109-156: The whole paragraph discussed very little about the specific mechanism of miRNA modulation. It focuses more on introduction of current research directions, if so, I suggest changing the title.
We thank the distinguished reviewer for the provided suggestions, we revised the title.
- Line 43-52: Please provide references.
We highly appreciate the observation in this regard and we thank again for the provided suggestions, we clarified this aspects in the manuscript.
- Line 89: I suggest changing subtitle, this is a reviewing paper not research paper.
We highly appreciate the observation in this regard and we thank again for the provided suggestions, we changed the subtitles.
- Line 108: Please unify the format of subtitles.
We highly appreciate the observation in this regard and we thank again for the provided suggestions, we revised this aspects in the manuscript.
- Line 217-224: Please combine with next paragraph.
We really appreciate the observation, we restructured the paper.
- Line 229-239: Please combine with next paragraph.
- Line 272-287: Why did author discuss the AMPs and FMT interaction under the subtitle of "Implications of miRNAs in IBD Pathogenesis"?
- Line 301-304: How about moving this to the end of section?
We highly appreciate the observation in this regard and we thank again for the provided suggestions, we clarified this aspects in the manuscript, and change the sections according to your suggestions. We thank you very much, your reviewwas very important for improving our manuscript.
Reviewer 2 Report
Comments and Suggestions for Authors
line 1 - is this review or communication?
title should be improved, do not use abbreviations
abstract needs to be rewritten, and English language improved. It is hard to read at this point.
line 32 - explain in depth vaccine comparison
references are usually in the sentence and before . so please check author guidelines
line 96 - [insert years]? this is just unprofessional at this point
Authors must proofread their English (native speaker), follow intructions for authors by journal and include all the relevant references when writing a review article.
Comments on the Quality of English Language
it is hard to understand some parts and sentences
This is poorly written manuscript and I do not believe it is appropriate for publication in highly respected journal such as Nutrients.
Author Response
Reviewer 2
line 1 - is this review or communication?
title should be improved, do not use abbreviations
abstract needs to be rewritten, and English language improved. It is hard to read at this point.
line 32 - explain in depth vaccine comparison
references are usually in the sentence and before . so please check author guidelines
line 96 - [insert years]? this is just unprofessional at this point
Authors must proofread their English (native speaker), follow intructions for authors by journal and include all the relevant references when writing a review article.
Comments on the Quality of English Language
it is hard to understand some parts and sentences
This is poorly written manuscript and I do not believe it is appropriate for publication in highly respected journal such as Nutrients.
We thank the distinguished reviewer for the provided suggestions, we revised the review according to your suggestions and reorganized the manuscript. We also reconsidered the comparison with a vaccine and rewrite the references according to the author guidelines.
Reviewer 3 Report
Comments and Suggestions for Authors
Brusnic et al. MicroRNAs, such as miR-21, miR-146a, and miR-155, play crucial roles in IBD pathogenesis and offer potential therapeutic targets, with microbiota and FMT influencing miRNA regulation for personalized treatment. Moreover, the result is technically sounded.
The following are some comments and suggestions that are given to improve the manuscript:
Comment 1: What are the specific interaction mechanisms between microbiota and miRNA regulation in IBD.
Comment 2: How regulation of miRNAs and AMPs by FMT impacts clinical outcomes in IBD patients.
Comment 3: Ask whether the regulation of miRNAs and AMPs will have a feedback effect on the gut microbiota.
Comment 4: It is recommended to evaluate the potential of miRNAs in IBD therapy, particularly how they modulate immune responses and inflammation by targeting miRNAs.
Author Response
Reviewer 3
Brusnic et al. MicroRNAs, such as miR-21, miR-146a, and miR-155, play crucial roles in IBD pathogenesis and offer potential therapeutic targets, with microbiota and FMT influencing miRNA regulation for personalized treatment. Moreover, the result is technically sounded.
The following are some comments and suggestions that are given to improve the manuscript:
Comment 1: What are the specific interaction mechanisms between microbiota and miRNA regulation in IBD.
We thank the distinguished reviewer for the provided observation, we explained in detail how microbiota influences the miRNA in the fallowing paragraph and also further in the manuscript.
”Microbes in the gut can produce miRNA-like molecules that may directly interact with the host’s cellular machinery, influencing gene expression.[40] FMT can induce epigenetic changes, including DNA methylation and histone modification, which can affect miRNA expression and thereby modulate gene regulation. [40, 61, ,72] The gut microbiota plays a crucial role in shaping the host’s immune system.[71,72] Changes in the microbial community through FMT can lead to the modulation of miRNAs that are involved in immune cell differentiation and function.[72] Clinical implications are emphasized due to changes in miRNA expression profiles after FMT, which can serve as biomarkers for treatment response, helping to predict which patients are likely to benefit from FMT.[5,6,7] Interaction between gut microbiota and miRNA expression highlights the potential for personalized medicine approaches in IBD, where treatments are tailored based on individual microbiota and miRNA profiles.[73] MiRNAs have multiple clinical benefits and are target of new research directions to elucidate the precise mechanisms by which FMT influences miRNA expression and the downstream effects on gene regulation.[11, 20-25]
Investigating the long-term effects of FMT on miRNA regulation and the stability of these changes over time is crucial for understanding its potential as a durable therapeutic intervention. Exploring the synergistic effects of FMT combined with other treatments, such as miRNA-based therapies or traditional IBD medications, could enhance therapeutic outcomes.[30-45]”
Comment 2: How regulation of miRNAs and AMPs by FMT impacts clinical outcomes in IBD patients.
We highly appreciate the distinguished reviewer’s observation with regard to the regulation of miRNA and AMPs, we discussed in detail in the paragraph regarding miRNA and AMPs.
We used table 3 to resume the interplay between miRNA and AMP s
”Table 3. How AMPs and miRNAs regulate immune responses, inflammation, and gut homeostasis.
|
AMP |
miRNA |
Interplay Mechanism |
Role in IBD |
|
LL-37 |
miR-21 [77] |
miR-21 modulates LL-37 expression, affecting inflammation levels |
High levels of miR-21 correlate with reduced AMP function, leading to elevated inflammation |
|
Defensins |
miR-146a [78] |
miR-146a controls the immune response and can upregulate defensins |
Defensins help maintain intestinal barrier, while miR-146a limits excessive inflammation |
|
HBD-2 |
miR-155 [78] |
miR-155 affects HBD-2 expression, enhancing inflammatory response when overexpressed |
Elevated miR-155 correlates with increased inflammation and impaired AMP response |
|
RegIIIγ |
miR-223 [79] |
miR-223 regulates neutrophil function and maintains gut barrier by modulating RegIIIγ |
Balances immune response, supporting AMP activity in intestinal integrity |
|
Cathelicidins |
miR-122 [77] |
miR-122 downregulation supports cathelicidin function in inflammation |
Cathelicidins protect against pathogens, while miR-122 modulation reduces fibrosis |
„
Comment 3: Ask whether the regulation of miRNAs and AMPs will have a feedback effect on the gut microbiota.
We highly appreciate the distinguished reviewer’s suggestion, we tried to cover this topic in the second paragraph.
Comment 4: It is recommended to evaluate the potential of miRNAs in IBD therapy, particularly how they modulate immune responses and inflammation by targeting miRNAs.
We highly appreciate the distinguished reviewer’s observation we discussed this topic in the third paragraph, regarding personalized treatment in IBD.
Round 2
Reviewer 1 Report
Comments and Suggestions for Authors
The revised manuscript has addressed all the comments, although I still suggest the author using the same format (bold, non-italic) for all subtitles.
Author Response
We thank the distinguished reviewer for the appreciation and provided suggestions, we revised the subtitles.
Reviewer 2 Report
Comments and Suggestions for Authors
The manuscript has been improved according to the comments and is now acceptable for publication
Author Response
We thank the distinguished reviewer for the appreciation, we highly appreciate your time and help to improve our paper.